# How the Material Characteristics of Optical Fibers and Soil Influence the Measurement Results of Distributed Acoustic Sensing

**DOI:** 10.3390/s23177340

**Published:** 2023-08-23

**Authors:** Ke Jiang, Lei Liang, Xiaoling Tong, Feiyu Zeng, Xiaolong Hu

**Affiliations:** National Engineering Research Center of Fiber Optic Sensing Technology and Networks, Wuhan University of Technology, Wuhan 430070, China; k.jiang@whut.edu.cn (K.J.); tongxl@whut.edu.cn (X.T.); 318575@whut.edu.cn (F.Z.); huxiaolong123@whut.edu.cn (X.H.)

**Keywords:** distributed acoustic sensing, fiber optic cable, sensitivity improvement, vibration response

## Abstract

Fiber optic distributed acoustic sensing (DAS) technology is widely used in security surveillance and geophysical survey applications. The response of the DAS system to external vibrations varies with different types of fiber optic cable connections. The mechanism of mutual influence between the cable’s characteristics and DAS measurement results remains unclear. This study proposed a dynamic model of the interaction between the optical cable and the soil, analyzed the impact of the dynamic parameters of the optical cable and soil on the sensitivity of the DAS system, and validated the theoretical analysis through experiments. The findings suggest that augmenting the cable’s bending stiffness 5.5-fold and increasing its unit mass 4.2-fold result in a discernible reduction of the system’s response to roughly 0.15 times of its initial magnitude. Cables with lower unit mass and bending stiffness are more sensitive to vibration signals. This research provides a foundation for optimizing vibration-enhanced fiber optic cables and broadening the potential usage scenarios for DAS systems.

## 1. Introduction

Fiber optic distributed acoustic sensing (DAS) systems utilize the interference changes of backscattered Rayleigh light in an optical fiber to demodulate external disturbance signals, enabling distributed vibration sensing on fiber optic cables [1], widely applied in geophysical surveys [2,3,4], security surveillance [5,6], and various other fields [7,8,9,10]. As bare fiber optics are fragile, they are generally encased in cables for practical engineering use. The encapsulation layer enhances the reliability of the optical fiber, but concurrently diminishes its sensitivity to environmental vibrations. Currently, commercial communication cables often serve as a sensor for DAS systems in many applications, the design principles of communication cables are focused on reducing fiber transmission loss and improving mechanical performance in it, which is contrary to the requirements for sensitivity to the external environment when used as a sensor. Moreover, as a measurement sensor that responds to seismic waves, the cable’s structure and physical properties, and its coupling with the surrounding environment, are closely related to the measurement sensitivity of the DAS system, but the mechanism of mutual influence between them remains unclear. Therefore, which parameters of fiber optic cables affect the sensitivity of a DAS system is the focus of this paper.

The signal-to-noise ratio (SNR) of a DAS system is a crucial parameter of the system’s performance. Various methods have been proposed to enhance the SNR of the system. These methods include modifications to the design of the optical fiber [10,11], updating the system hardware [12,13], and digital signal processing methods [14,15,16]. Furthermore, as a sensor of the DAS system, the internal structure, mechanical properties, and environmental coupling of the optical cable will all affect the SNR of the system.

In an interaction model between optical fiber cables and soil, Kuvshinov [17] assumes the optical fiber cable to be a uniform elastic medium and derives the relationship between the optical fiber strain and the soil strain based on the Navier equations and stress–strain relationships for both fully coupled and sliding cables. The study analyzes the impact of primary waves, shear waves, and Rayleigh waves on the optical fiber cable and discusses the selection of appropriate optical fiber winding angles to enhance the sensitivity of distributed acoustic sensing systems. References [18,19,20] mention that the method of using the elastic foundation beam can be applied to analyze the dynamic displacement or strain response of buried pipelines during earthquakes. This method can be applied as an analogy to study the interaction between optical fiber cables and the surrounding soil.

How to select or design the most suitable optical cable for a specific application scenario is currently a hot research topic. The common approach is to connect diverse kinds of optical cables to a DAS system and place them in their respective application scenarios, gauge the disparities in their response to the target signal, and pick the optimal optical cable from among them [21,22,23]. However, this method can only discover comparatively superior choices from current communication optical cables and is not a purpose-built, custom sensing cable. Customized sensing cables have the potential to achieve a sensitivity several times greater than that of communication optical cables. Han et al. [24] proposed a sensitivity-enhanced cable that is 10 times higher in acoustic sensitivity than a rigid cable with armor reinforcement. AP Sensing, Corning [25], and other optical cable manufacturing companies offer the customization of sensing optical cables to achieve a balance between cable thermal conductivity, ruggedness, and costs. Some customized optical cables used in various fields are shown in Table 1.

When utilizing buried fiber optic cables for seismic wave sensing, the stranding angle of the fiber within the cable has a significant impact on the sensitivity of DAS systems. Helically coiled fiber cables result in greater seismic wave responsiveness in comparison to straight cables [26]. Incorporating a densely coiled fiber optic cable can increase its sensitivity to environmental vibrations by increasing the total length of the helically wound fibers within the cable [27]. Seismic wave incidence angle, medium-to-cable coupling degree, and the medium’s inherent properties and other factors all impact the outcomes of measuring seismic waves using fiber cables [28]. The present research results are predominantly focused on local fiber cable characteristics and their effect on measurements, with scant literature on the impact of the overall performance parameters of the fiber cable on vibration measurement sensitivity.

This paper aims to address the seismic-wave-sensing scenario of buried optical fibers, treating the optical fibers and soil as a dynamic system, and to investigate the connection between system parameters and DAS sensitivity. Subsequently, a series of field tests were conducted to validate the effectiveness of the proposed approach. This study offers a perspective and theoretical guidance for the design of vibration-sensitive optical cables.

## 2. The Kinetic Model of Buried Optical Cables

To build a kinetic model for a buried optical cable, it is necessary to make certain assumptions and simplifications. Firstly, it is assumed that the cable has a large aspect ratio and oscillates slightly in its equilibrium position, with transverse vibration being the main form of deformation and axial vibration being disregarded. Secondly, based on the method of cable laying, the cable is simplified as a simple supported string with bending stiffness and internal tensile, while the soil is simplified as a spring and damper system [17,18,19], as shown in Figure 1.

The bending stiffness of the cable is *EI*, its unit mass is *m*, and *l* and *T* are the length and internal tensile of the optical cable, respectively. *K* and *C* represent the equivalent stiffness and damping of soil, respectively. The lateral load *p*(*x*, *t*) and lateral displacement *y*(*x*, *t*) are both dependent on position and time. The force condition of the fiber optic microelement with a length of dx is depicted in Figure 1. According to the D’Alembert principle, the dynamic balance equation and moment balance equation of the microelement in the vertical direction are listed as:(1)$$Q+pdx-(Q+\frac{\partial Q}{\partial x})-m\frac{\partial^{2}y(x,t)}{\partial t^{2}}dx-C\frac{\partial y(x,t)}{\partial t}-Ky(x,t)=0.$$
(2)$$M+Qdx-T\frac{\partial y}{\partial x}dx-(M+\frac{\partial M}{\partial x}dx)=0.$$
where *Q* represents the shear force in the optical cable and *M* represents the moment in the optical cable. By merging Equations (1) and (2) and substituting the correlation between moment and curvature $M=EI\partial^{2}y/\partial x^{2}$, we obtain the dynamic equation of the system:(3)$$EI\frac{\partial^{4}y(x,t)}{\partial x^{4}}+T\frac{\partial^{2}y(x,t)}{\partial x^{2}}+m\frac{\partial^{2}y(x,t)}{\partial t^{2}}+C\frac{\partial y(x,t)}{\partial t}+Ky(x,t)=p(x,t).$$

The displacement response of an optical cable under external excitation can be derived by resolving the kinetic equation, from which the strain response of the cable can be deduced. The strain response of the cable is proportionate to the sensitivity of the DAS system. The process of solving the kinetic equation is divided into two primary steps: Firstly, utilizing the method of separation of variables to determine the system’s natural frequency and mode shapes. Secondly, applying the modal superposition method, in conjunction with the Duhamel integral, to determine the system’s response.

When determining the natural frequency and mode shapes of the system, we may temporarily disregard the damping coefficient and loading term in Equation (1), and they can be obtained as follows:(4)$$EI\frac{\partial^{4}y(x,t)}{\partial x^{4}}+T\frac{\partial^{2}y(x,t)}{\partial x^{2}}+m\frac{\partial^{2}y(x,t)}{\partial t^{2}}+Ky(x,t)=0.$$

This is a linear homogeneous partial differential equation with constant coefficients, which is solved by the method of separation of variables, and the solution is:(5)$$\omega_{j}=\sqrt{\frac{EI}{m}\frac{j^{4}\pi^{4}}{l^{4}}+\frac{T}{m}\frac{j^{2}\pi^{2}}{l^{2}}+\frac{K_{t}}{m}}(j=1,2,\cdots ).$$
(6)$$\phi_{j}(x)=\sin\frac{j\pi }{l}x\;(j=1,2,\cdots ).$$
where *j* represents the mode number of the system and *ω_j_* and *ϕ_j_*(*x*) represent the natural frequency and mode shapes of the system, respectively.

We can utilize the method of mode shape superposition to determine the system’s response after obtaining the natural frequency and mode shapes of the system as follows:(7)$$y(x,t)=\overset{\infty }{\underset{j=1}{\sum }}\phi_{j}(x)q_{j}(t).$$
where *q_j_*(*t*) is the displacement response of the jth order mode shape throughout time t. The system’s mode shapes are obtained, and the displacement response of the system can be ascertained upon obtaining *q_j_*(*t*). By substituting Equation (7) into Equation (1), pre-multiplying by *ϕ_j_*(*x*), and integrating over the length of the optical cable, we can obtain:(8)$$\begin{array}{l} \int_{0}^{l}\frac{d^{2}}{dx^{2}}\left[EI\frac{d^{2}\phi_{j}(x)}{dx^{2}}\right]\phi_{j}(x)dxq_{j}(t)+\int_{0}^{l}K\phi_{j}^{2}(x)dxq_{j}(t)+\int_{0}^{l}m\phi_{j}^{2}(x)dx{\ddot{q}}_{j}(t)+\int_{0}^{l}T\frac{d^{2}\phi_{j}(x)}{dx^{2}}\phi_{j}(x)dxq_{j}(t)\\ +\overset{\infty }{\underset{i=1}{\sum }}\int_{0}^{l}C\phi_{j}^{2}(x)dx{\dot{q}}_{j}(t)=\int_{0}^{l}\phi_{j}(x)p(x,t)dx. \end{array}$$

Utilizing orthogonality property of the mode shapes:(9)$$\frac{d^{2}}{dx^{2}}[EI\frac{d^{2}\phi_{j}}{dx^{2}}]+T\frac{d^{2}\phi_{j}}{dx^{2}}+K\frac{d\phi_{j}}{dx}=\omega_{j}{}^{2}m\phi_{j}(x).$$

Introducing the concept of generalized mass $M_{j}=\int_{0}^{l}m\phi_{j}^{2}(x)dx$, generalized load $P_{j}(t)=\int_{0}^{l}\phi_{j}(x)p(x,t)dx$, and damping parameters $\zeta_{j}=C/2m\omega_{j}$ to simplify Equation (8), we can obtain:(10)$${\ddot{q}}_{j}(t)+2\zeta_{j}\lambda_{j}{\dot{q}}_{j}+\omega_{j}^{2}q_{j}(t)=P_{j}(t)/M_{j}\;(j=1,2,\cdots ).$$

Equation (10) is a synthesis of an infinite number of single-degree-of-freedom vibration equations, from which the final resolution can be attained by summing the outcomes of each mode. For any external load, the dynamic response of the system under zero initial conditions can be resolved through the utilization of the Duhamel integral method. If $\omega_{D}=\omega_{j}\sqrt{1-\zeta_{j}{}^{2}}$, the form of the solution is
(11)$$q_{j}(t)=\frac{1}{M_{j}\omega_{D}}\int_{0}^{t}P_{j}(\tau )e^{-\zeta_{j}\omega_{j}(t-\tau )}\sin\omega_{D}(t-\tau )d\tau .$$

If the external load is a periodic signal that is easily integrable, it can be obtained via the solution of indefinite integrals. Conversely, if not directly integrable, it can be attained through approximate integration after a transformation.
(12)sin(wt−wτ)=sinwtcoswτ−coswtsinwτ.

Then, Equation (11) can be transformed into
(13)$$q_{j}(t)=\frac{1}{M_{j}\omega_{D}}[A(t)\sin\omega_{D}t-B(t)\cos\omega_{D}t].$$
(14)$$A(t)=\int_{0}^{t}P_{j}(\tau )e^{-\zeta_{j}\omega_{j}(t-\tau )}\cos\omega_{D}(\tau )d\tau .$$
(15)$$B(t)=\int_{0}^{t}P_{j}(\tau )e^{-\zeta_{j}\omega_{j}(t-\tau )}\sin\omega_{D}(\tau )d\tau .$$

The approximate integral equation of *A*(*t*) using Simpson’s rule is:(16)$$A(t)=A(t-2)+\frac{\Delta \tau }{3}(y(t-2)+4y(t-1)+y(t)).$$
where $y(t)=P_{j}(t)e^{-\zeta_{j}\omega_{j}t}\cos\omega_{D}(t)$. *B*(*t*) can be obtained with the same method. 

Upon obtaining *q_j_*(*t*), the system’s response can be obtained using Equation (7). Within the elastic range, utilizing the relationship between bending strain and curvature, the bending strain of the optical fiber can be obtained as follows:(17)$$\varepsilon (x,t)=\frac{D}{2}\overset{\infty }{\underset{j=1}{\sum }}\frac{\partial^{2}\phi_{j}(x)}{\partial x^{2}}q_{j}(t).$$
where *D* is the diameter of the cable.

The DAS system senses external vibrations through detecting changes in the optical phase in the fiber. Its sensitivity can be represented by the magnitude of phase change.
(18)$$\Delta \varphi =\beta L\varepsilon \left\{1-\frac{1}{2}n^{2}\left[(1-v)P_{12}-vP_{11}\right]\right\}.$$
where Δ*φ* represents the phase change of light waves in fiber optics, *L* represents the length of the fiber optics, *β* represents the constant of propagation of light waves in fiber optics, *ε* is the longitudinal strain of the fiber, *n* represents the fiber refraction index, *v* is Poisson’s ratio, and *P*_11_ and *P*_12_ are the elastic-optic coefficients. Equation (18) indicates a proportional relationship between the fiber strain change and fiber phase change. So, the strain response of buried optical fiber cable can serve as the sensitivity of the DAS system.

## 3. Analysis of Parameters Affecting DAS Sensitivity

The primary objective of this section is to investigate the relationship between the strain response of the optical fiber and system parameters such as the bending stiffness, linear density, and length of the cable. By utilizing the control variable method, we analyze the impact of various parameters on the strain response of the optical fiber. The external load is represented as *p*(*x, t*) = sin*(wt)*, *w* represents the vibration frequency of the external load, and *t* represents the duration of the external load. The typical order of magnitude for each parameter is shown in Table 2.

Figure 2 shows the strain time–history curve of the midpoint in the cable under typical magnitude. The response was evaluated by utilizing the first 30 modal orders of the system, with a computed time interval of 0.0001 s. The signal manifests a sinusoidal trend of 50 Hz in its entirety, with fluctuations arising from the coupling of higher mode shapes and lower mode shapes. Due to the presence of fluctuations in the signal, the average peak-to-peak value of the system’s strain response over 0.5 s was utilized as a substitute for the sensitivity of the DAS system.

Figure 3 shows the average peak-to-peak strain response in the mid-section of the optical fiber under varying sinusoidal loads at different frequencies. The peak position in each subfigure corresponds to the system’s first-order natural frequency. For the vibration sensing system, the first 1/10th of the natural frequency constitutes the measurement range, and the corresponding strain of the optical fiber within this range represents the system’s sensitivity.

Due to the rapid attenuation of high-frequency seismic waves in soil, the frequency of seismic waves that propagate in soil generally falls below 200 Hz. The enlarged detail in Figure 3 displays the system’s strain response within the range of 200–1000 rad/s (20–200 Hz). Table 3 shows the strain simulation values of three types of optical cables in Figure 3 at 200 Hz. Figure 3a clearly depicts that as soil stiffness increases, system sensitivity decreases and its impact on system sensitivity gradually decreases. When soil stiffness increases 3-fold, sensitivity decreases to 0.35 times its initial value. Figure 3b shows that as the unit mass of the optical cable increases, system sensitivity decreases and its impact on system sensitivity gradually decreases. When the unit mass of the optical fiber is tripled, the sensitivity varies by 0.58 times. Figure 3c indicates that as the bending stiffness of the cable increases, the inherent frequency of the system remains unchanged, yet its sensitivity decreases. When the bending stiffness of the cable changes 3-fold, the sensitivity varies by 0.84 times. Figure 3d reveals that the impact of cable length on system sensitivity follows a similar pattern to that depicted in Figure 3, yet exerts a greater influence on the system’s sensitivity; when the length of the cable changes 3 times, the sensitivity decreases to 0.08 times its initial value. Figure 3e,f reveal that as the tensile of the optical cable and soil damping increase, the system’s natural frequency remains unchanged, as does its sensitivity.

To sum up, the soil stiffness, the unit mass of the cable, the length of the cable, and its bending stiffness have a significant impact on the system’s sensitivity, whereas soil damping and cable tensile have a relatively minor impact on the system’s sensitivity.

## 4. Experimental Measurements and Discussion

In this section, the effect of soil stiffness, unit mass of the optical cable, cable length, and its bending stiffness on the system sensitivity are explored through experiments. Soils with three different levels of confining pressure were used to simulate the differentiation of soil stiffness [29], while the use of three different materials for the cable allowed for significant differences in its mass and bend stiffness. Two lengths of cable were utilized to study the impact of cable length on system response. The experimental setup, as depicted in Figure 4, entailed the installation of three test optical cables within a soil trench. Periodic vibration signals are generated by an exciter and transmitted to the test cables through the underlying stratum and refilled sand.

### 4.1. Optical Fiber Cable

Three cable types were selected for the experiments: a communication cable, GYTA53, and two simulated cables, made from aluminum pipes and thermoplastic polyurethane (TPU) tubes, respectively, to create contrasts in mass and stiffness. The manufacturer of GYTA53 is Yangtze Optical Fibre and Cable Joint Stock Limited Company (YOFC). The density and elastic modulus of aluminum and TPU are 2.72 g/cm^3^ and 79 Gpa, and 1.17 g/cm^3^ and 32 Mpa, respectively. The parameters of the three cables can be found in Table 4.

This study focuses on the strain response of the entire cable to prevent measurement errors caused by differences in the internal structure of the cable; a 0.9 mm tight-buffered cable was fixed to the outer surface of three test cables, employing the utilization of heat-shrink tubing shown in Figure 4b. The response of the tight-buffered cable serves as the three cables’ response to external excitation.

The tight-buffered cable features a fusionless design to deter the formation of additional reflection surfaces in the optical fibers, thus eliminating the loss of back-reflected Rayleigh scattering signals and avoiding any potential impact on measurement results. As shown in Figure 4a, each test cable was approximately 10–15 m in length. A 0.9 mm tight-buffered cable was affixed to the outer surface of three test cables, leaving a stretch of optical fiber loops ranging from 120 to 150 m between each cable to isolate the measurement signals from interference. The overall length of the cable, including the fiber optic loop at the beginning and end, was approximately 500 m. The tight-buffered cable was led from cable 1 to the DAS system. The testing site is illustrated in Figure 4c. Figure 4d shows the cross-section diagrams of the three types of optical cables. The DAS system, developed by the National Engineering Research Center of Fiber Optic Sensing Technology and Networks at Wuhan University of Technology, was utilized to demodulate the vibrational signals generated by the optical cable. The phase demodulation technique of the DAS is based on a 3 × 3 coupler. The pulse width of the DAS system is 50 ns, the optical source has a power of 20 mW, the sampling rate is 20,000 Hz, the gauge length is 5 m, and the spatial resolution is 1 m.

### 4.2. Test Procedure

The test procedure is depicted in Table 5.

Test 1 aimed to investigate the response of cables under different soil stiffness, with cables being buried to the same length. Test 2 examined the response of cables with various buried lengths under a uniform soil stiffness, and a common excitation source. The vibration source is an eccentric exciter with a power of 30 W and a maximum excitation force of 200 N. The rotational speed is 3000 r/min, and the vibration frequency can be turned from 0 to 60 Hz. The utilization of a controllable and stable excitation source facilitates subsequent signal processing and analysis. The eccentric vibrator is coupled to the ground through a foot bolt base and is positioned at a vertical testing trench of 2 m.

### 4.3. Discussion

Figure 5 presents a waterfall chart of DAS test signals for a cable buried with a length of 10 m under soil stiffness *K*_1_. The *x*-axis represents the number of channels in the DAS system, symbolizing the distance between the cable and the DAS device. The cables 1, 2, and 3 correspond to the channel ranges of 21–31, 182–192, and 313–323, respectively, representing the length of the 10 m cables. The center positions of cables 1, 2, and 3 are represented by the 25th, 186th, and 317th channel, respectively. The *y*-axis of the waterfall chart corresponds to the measurement time of the DAS system.

In Figure 5, the eccentric vibrator was activated at 0.8 s, and the vibrator vibrations tended to be stable at 4 s. The subsequent analysis is based on the stable section signals. The depth of the waterfall chart’s color represents the strength of the vibration, with deeper colors indicating stronger vibrations. The chart shows that cable 2 has the most intense vibration, while cable 3 has the weakest.

Figure 6 presents the time-domain data and its frequency-domain transformation of the 25th, 186th, and 317th channels of the DAS system from 6 s to 6.5 s. The data underwent a 10–200 Hz bandpass filtering to eliminate noise interference. The *y*-axis in the time-domain graph pertains to the demodulation optical phase value. The time-domain graph illustrates that, at the same location of the three cables, the amplitude of the signal measured by cable 2 is approximately 1.5 times that of optical cable 1 and three times that of optical cable 3. The signals in the time domain exhibit some degree of fluctuation, which can be attributed to the incomplete consolidation of cable and soil, and the resulting unstable phase difference between the two vibrating elements leads to unsteady signal characteristics. The signal primarily consists of a sinusoidal signal with a fundamental frequency of 50 Hz, but fluctuations in the signal can also arise from the 100 Hz harmonic frequency generated by the coupling between the exciter and soil.

Figure 7 displays the results of Test 1, where each data point represents the average peak-to-peak value of the 0.5 s phase data. Ten random data points, ranging from 4–9 s, were taken as the result of a single measurement. The bar chart corresponds to the mean value of these ten data points. Figure 7 reveals that for the same cable, a higher soil stiffness results in smaller system responses, and for the same soil stiffness, a higher unit mass and bending stiffness of the cable results in smaller system responses. This is consistent with theoretical analysis. When fiber optic cable 1 serves as the sensor, the average phase change of the system corresponds to 4.16 × 10^−2^ rad, 3.06 × 10^−2^ rad, and 2.74 × 10^−2^ rad for the three soil stiffness levels. Cable 2 and cable 3 exhibit average phase changes of 11.84 × 10^−2^ rad, 10.56 × 10^−2^ rad, and 8.09 × 10^−2^ rad; and 1.72 × 10^−2^ rad, 1.61 × 10^−2^ rad, and 1.26 × 10^−2^ rad, respectively. When the soil stiffness increases two-fold, sensitivity decreases to approximately 0.66 times its initial value. When the cable bending stiffness increased by 5.5-fold, and its unit mass increased by 4.2-fold, the system response was observed to have declined to approximately 0.15 times the original value.

As indicated by Figure 8, the magnitude of system response decreases with an increase in cable buried length, which is in accordance with the theoretical analysis inference. When fiber optic cable 1 serves as the sensor, the average phase change of the system corresponds to 4.16 × 10^−2^ rad, 2.86 × 10^−2^ rad for the two buried lengths of cable. Cable 2 and cable 3 exhibit average phase changes of 11.84 × 10^−2^ rad and 3.96 × 10^−2^ rad; and 1.72 × 10^−2^ rad and 0.94 × 10^−2^ rad, respectively. When the cable buried length increases 1.5-fold, sensitivity decreases to approximately 0.33 times its initial value.

The test results indicate that cables with lower unit mass and bending stiffness are more sensitive to vibration signals. Commercial optical cables are typically equipped with metal armor or loose tube structures to minimize stress on the optical fibers, ensuring their longevity. However, this may lead to a decrease in the fibers’ sensitivity. The increased use of metal components in the cables results in higher unit mass and bending stiffness, consequently reducing the sensitivity of the optical fibers.

There are typically two methods for laying optical cables within soil: direct burial and duct installation. This article only discusses the scenario of direct burial. The measurements from reference [25] indicate that the measured signal when directly buried is approximately 10 dB higher compared to duct installation. We will also delve into the dynamic model of cable deployment within duct and analyze the influencing factors. Additionally, the sensor cables designed in references [24,25] all share the characteristics of being lightweight, having low bending stiffness, and tight fitting. This further validates the reliability of the findings presented in this article.

## 5. Conclusions

The DAS system sensitivity may fluctuate several times due to the influence of the optical cables and their layout methods. We proposed a dynamic model of the interaction between the optical cable and the soil, and investigated the relationship between cable mechanical characteristics and DAS sensitivity. The theoretical results indicated that soil stiffness, cable bending stiffness, and cable unit mass are crucial factors that impact the sensitivity of the system. The experimental results indicated that augmenting the cable’s bending stiffness 5.5-fold and increasing its unit mass 4.2-fold result in a discernible reduction of the system’s response to roughly 0.15 times its initial magnitude. Cables with lower unit mass and bending stiffness are more sensitive to vibration signals. The present study investigates the impact of cable parameters on the system sensitivity when the cable is considered as an indivisible entity. Future studies will investigate the influence of the internal structure of the optical cable, such as the stretching window, twisting angle, and grease filling of optical units, as well as the inner structural design of reinforcing elements, on the sensitivity of the system, creating a design system for sensing cables and furnishing a targeted design basis for the application of sensing cables in various fields.

## Figures and Tables

**Figure 1 sensors-23-07340-f001:**
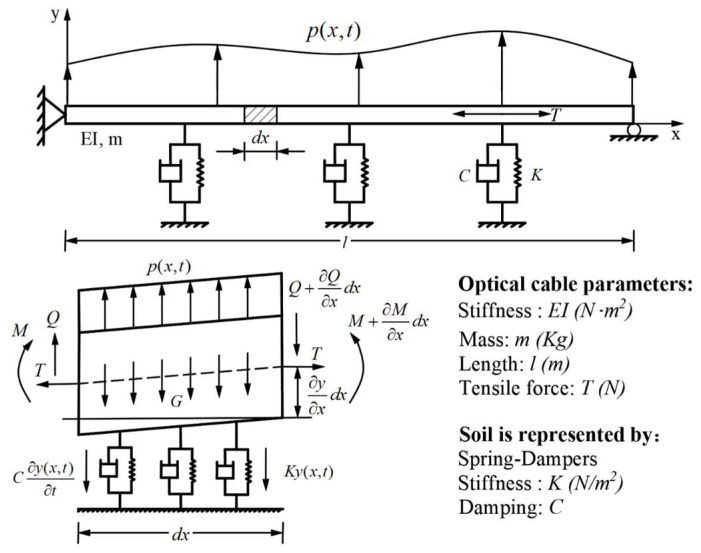
The kinetic model of the buried optical cable and the forces acting on the microelements of the optical cable.

**Figure 2 sensors-23-07340-f002:**
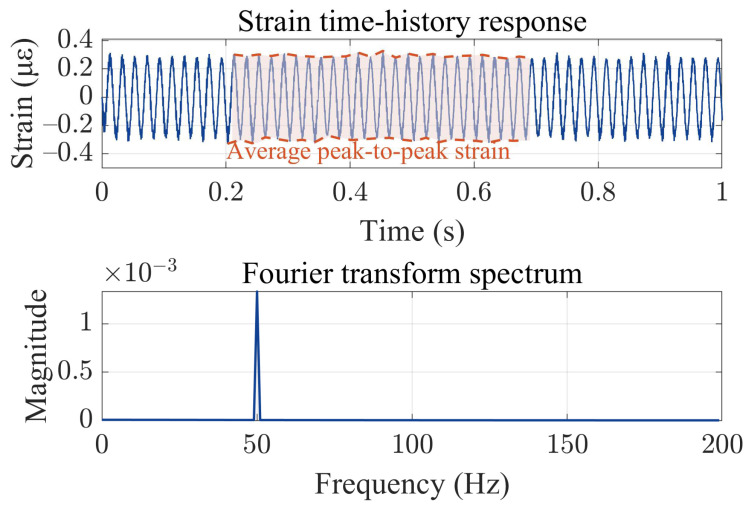
The strain time-history response and Fourier transform spectrum of the midpoint of the optical cable under a 50 Hz uniform sinusoidal load.

**Figure 3 sensors-23-07340-f003:**
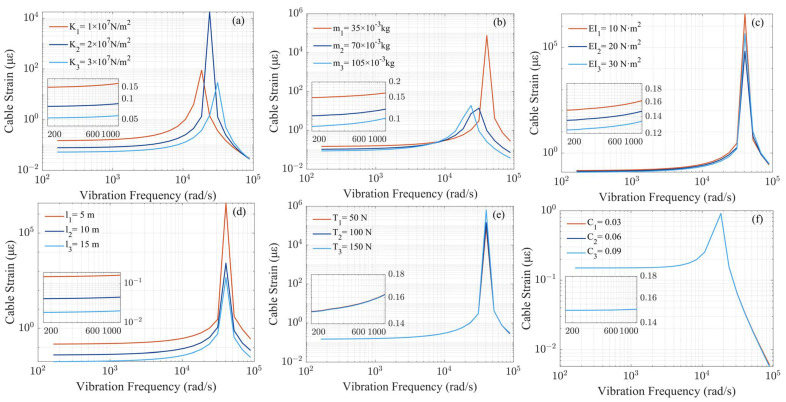
The relationship between soil stiffness (**a**), unit mass of the cable (**b**), bending stiffness of the cable (**c**), cable length (**d**), tensile of the cable (**e**), soil damping (**f**), and average peak-to-peak strain response in the mid-section of the optical cable under different sinusoidal load frequencies.

**Figure 4 sensors-23-07340-f004:**
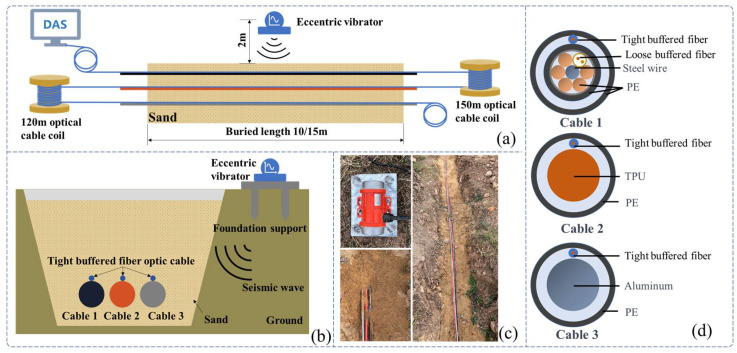
The experimental setup. (**a**) The overall diagram of the testing system, (**b**) the layout of the vibration source and optical cable, (**c**) the testing site, and (**d**) a cross-section of the three types of cables.

**Figure 5 sensors-23-07340-f005:**
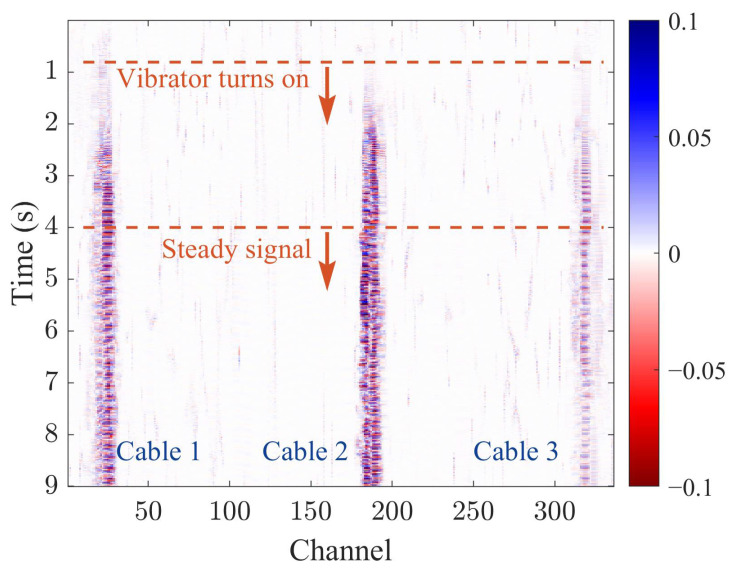
The waterfall chart of DAS test signals for a cable buried with a length of 10 m under soil stiffness *K*_1_.

**Figure 6 sensors-23-07340-f006:**
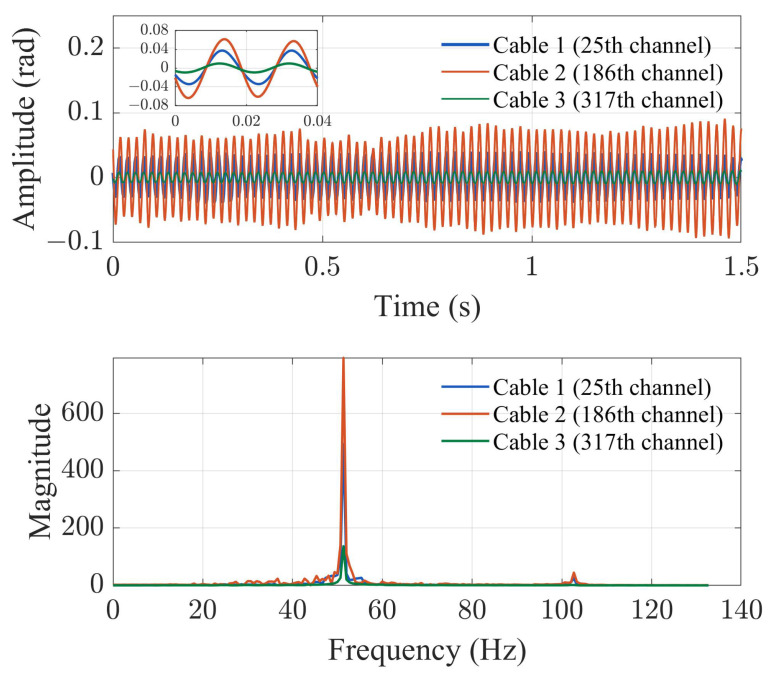
The time-domain phase amplitude and Fourier transform spectrum of the 25th, 186th, and 317th channels of the DAS system from 6 s to 6.5 s.

**Figure 7 sensors-23-07340-f007:**
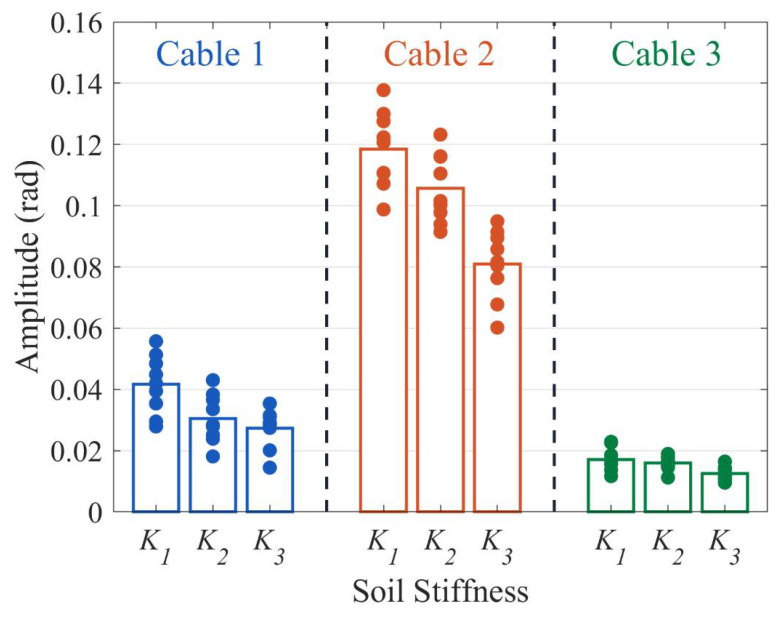
The phase response of three cables under three soil stiffnesses, with cables being buried 10 m.

**Figure 8 sensors-23-07340-f008:**
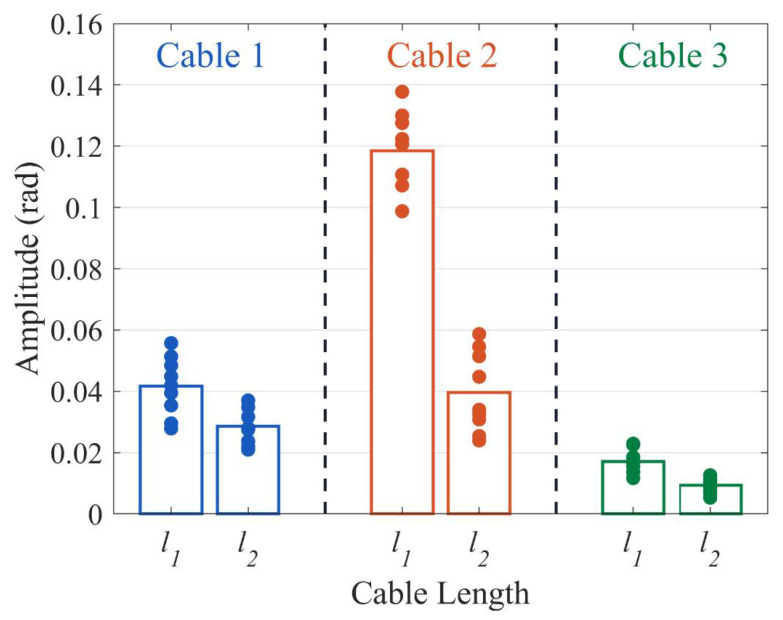
The phase response of three cables with two buried lengths under soil stiffness *K*_1_.

**Table 1 sensors-23-07340-t001:** Customized optical cables in various fields.

References	Application	Cross-Section
Daley [4]	Oil industry, downhole	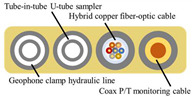
Yamasaki [10]	Oil industry, downhole	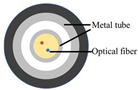
Hofmann [21]	Electric power industry	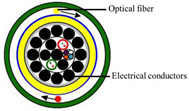
Hofmann [21]	Security surveillance, intrusion detection	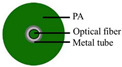
Han [23]	Geophysical survey	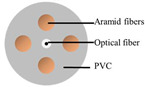
Freeland [25]	Geophysical survey	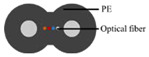

**Table 2 sensors-23-07340-t002:** System parameters and typical magnitude.

Symbol	Physical Meaning	Typical Magnitude
*K*	Stiffness of soil	~1 × 10^7^ N/m^2^
*C*	Damping of soil	~0.03
*m*	Unit mass of the optical cable	~35 × 10^−3^ kg/m
*EI*	Bending stiffness of the optical cable	~10 N·m^2^
*l*	Length of the optical cable	~5 m
*T*	Tensile of the optical cable	~50 N
*w*	Vibration frequency of the load	~50 Hz

**Table 3 sensors-23-07340-t003:** Strain simulation values of three types of optical cables in Figure 3 at 200 Hz.

	Cable 1 Strain (με)	Cable 2 Strain (με)	Cable 3 Strain (με)
(a)	0.149	0.078	0.053
(b)	0.149	0.106	0.087
(c)	0.149	0.135	0.124
(d)	0.149	0.041	0.018
(e)	0.149	0.148	0.148
(f)	0.149	0.149	0.149

**Table 4 sensors-23-07340-t004:** System parameters and typical magnitude.

	Unit Mass (kg/m)	Bending Stiffness (N·m^2^)	Outer Diameter (mm)
Cable 1	70.9 × 10^−3^	0.5	8.0
Cable 2	33.0 × 10^−3^	0.08	8.0
Cable 3	137.6 × 10^−3^	17.1	8.5

**Table 5 sensors-23-07340-t005:** System parameters and typical magnitude.

	Buried Lengths of Cables	Stiffness of Soil [29]
Test 1	*l*_1_ = 10 m	*K_1_* ≈ 1.4·10^7^ N/m^2^ (Confining pressure = 1 kPa) *K_2_* ≈ 2.0·10^7^ N/m^2^ (Confining pressure = 2 kPa) *K_3_* ≈ 2.8·10^7^ N/m^2^ (Confining pressure = 4 kPa)
Test 2	*l*_1_ = 10 m *l*_2_ = 15 m	*K_1_* ≈ 1·10^7^ N/m^2^

## Data Availability

The data that support the findings of this study are available from the corresponding author upon reasonable request.

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
