# Peer review of "How the Material Characteristics of Optical Fibers and Soil Influence the Measurement Results of Distributed Acoustic Sensing"

_sensors, 2023, doi:10.3390/s23177340_

Round 1

Reviewer 1 Report

I have several concerns regarding the article "Research on the Vibration Response of Fiber Optic Cable in Soil for DAS Measurement." Firstly, the article needs more clarity and coherence in its research methodology and results presentation. The article only briefly mentions a few studies without critically analyzing their methodologies, results, and limitations.

To improve the quality of the article, the authors should consider the following constructive suggestions:

1. Three cable types were selected for the experiments, but the authors still need to present more information about these cables. Considering the paper's objective, “a foundation for optimizing vibration-enhanced fiber optic cables and broadening the potential usage scenarios for DAS systems” the authors need to describe in detail the cable constructions used in the experiments.

2. The characteristics/specifications of the eccentric vibrator (power, torque, and impact) need to be presented.

3. Line 234, verify the value of velocity. Is it correct  3,000 r/s?

4. The authors need present the DAS system specifications, such as a pulse width, gauge length, and sampling rate.

Due to the falt of cable information, the discussion about the results is compromised.

Author Response

Response to Reviewer 1 Comments

I have several concerns regarding the article "Research on the Vibration Response of Fiber Optic Cable in Soil for DAS Measurement." Firstly, the article needs more clarity and coherence in its research methodology and results presentation. The article only briefly mentions a few studies without critically analyzing their methodologies, results, and limitations.

The title of the article should be changed to "How the Material Characteristics of Optical Fibers and Soil Influence the Measurement Results of DAS."

Point 1: Three cable types were selected for the experiments, but the authors still need to present more information about these cables. Considering the paper's objective, “a foundation for optimizing vibration-enhanced fiber optic cables and broadening the potential usage scenarios for DAS systems” the authors need to describe in detail the cable constructions used in the experiments.

Response 1: The cross-sectional diagrams of the three types of optical cables have been added to Figure 4.

Point 2: The characteristics/specifications of the eccentric vibrator (power, torque, and impact) need to be presented.

Response 2: The vibration source is an eccentric exciter with a power of 30W and a maximum excitation force of 200N. The rotational speed is 3000r/min and the vibration frequency can be turned from 0 to 60 Hz. The relevant content in the paper have been revised accordingly.

Point 3: Line 234, verify the value of velocity. Is it correct 3,000 r/s?.

Response 3: This is an error in the paper. The rotational velocity of the vibration source should be 3000 r/min. The relevant content in the paper have been revised accordingly.

Point 4: The authors need present the DAS system specifications, such as a pulse width, gauge length, and sampling rate.

Response 4: The pulse width of the DAS system is 50ns, the sampling rate is 20000Hz, and the gauge length is 5m. The relevant content in the paper have been revised accordingly.

Reviewer 2 Report

The manuscript proposed two parts of their work for analyzing the DAS system between the optical fiber cable and the surrounding soil/earth. Firstly, a numerical analysis using a kinetic model examines the relationship between unit mass, bending stiffness, cable length, cable tensile, soil damping, etc. of an optical fiber model. Secondly, three cables were investigated. The common GYTA53 type of optical communication cable, as well as two customed cables, were buried into the ground for testing. The testing signal was generated by a 50 Hz eccentric vibrator from the surfaces. Selected assumptions, scope, and setup of the test methods were provided in the manuscript.

Major comments for the authors to address:

1) Line 2 title, line 8 abstract, and line 294 conclusion should be rewritten.

2) The title does not describe the work done by the manuscript. It is too generalized while the manuscript is very narrow for a particular optical cable mode, soil model, excited vibration, environmental factors, etc test case.

3) The abstract does not provide readers with critical inputs and outcome numeric values. For example, the manuscript's main body examined too little of the cable structure to be included in the abstract.

4) The conclusion is not supported by the test setup as it is very narrowly focused on selected test assumptions and parameters.

5) Line 48, there are many commercial custom sensing cables designed for DAS's vibration sensing globally. It is advised for the authors to conduct a more vigorous background study on the type of cable used and use the same or similar structural design for the literature review and the main body of the manuscript. 

6) Line 75, is the "Kinetic model of Buried Optical Cable" cited from published work or newly proposed in this manuscript? Additional references should be added to support the theorem/ laws.

7) Are the assumptions sufficient to model practical seismic waves? Can the authors provide information on such as seismic waves as a use case in the manuscript with reference(s)? If not, the proposed D’Alembert principle, the dynamic balance equation, and the moment balance equation method may be not technically sound for modeling any optical cable.

8) Since the main bulk of the manuscript is on models, the background review should also include past publications on theoretical models. 

9) Line 152, can the authors kindly provide information on how the typical magnitude values are obtained? Is it from published work or from experiences/ internal data by the authors?

10) Line 152, 200, and Section 3 Analysis of parameters affecting DAS sensitivity, a periodic 50 Hz vibration are too simple for analysis for investigating DAS nowadays. Is suggested to use realistic periodic signals for the targeted seismic wave of interest. 

11) Line 190, a sum-up table will be useful for readers to glance through the data and results efficiently. 

12) Line 204, the cross-sectional detailed layout of the optical cables should be depicted in the manuscript. The manufacturer, model information, and specifications of the cables should be provided in the manuscript for a high-quality manuscript. 

13) Line 208, it is unclear what is the main material in Table 1. As a user of the GYTA52 cable myself, more information on the setup should be provided and justified as to why the current communication optical cable is not practical for seismic wave sensing.

14) Is the 8.00 mm outer diameter cable specification too thin and not practical for buried seismic wave sensing? There is no other structural information inside the manuscript to determine the mechanical properties of the proposed cables' setup. Even the types of sand and how it is set up in the pit will influence the results for example. 

15) Figure 6. Cable 2 and 3 seem to have stable amplitudes while cable 1 varies a bit. Are there other times when there are large variations? Any reason for the variations?

I hope my comments will improve the authors' manuscript and hopefully gain more impactful citations when published. 

Author Response

Response to Reviewer 2 Comments

The manuscript proposed two parts of their work for analyzing the DAS system between the optical fiber cable and the surrounding soil/earth. Firstly, a numerical analysis using a kinetic model examines the relationship between unit mass, bending stiffness, cable length, cable tensile, soil damping, etc. of an optical fiber model. Secondly, three cables were investigated. The common GYTA53 type of optical communication cable, as well as two customed cables, were buried into the ground for testing. The testing signal was generated by a 50 Hz eccentric vibrator from the surfaces. Selected assumptions, scope, and setup of the test methods were provided in the manuscript.

Point 1: Line 2 title, line 8 abstract, and line 294 conclusion should be rewritten.

Response 1: The title, abstract, and conclusion of the paper has been revised.

Point 2: The title does not describe the work done by the manuscript. It is too generalized while the manuscript is very narrow for a particular optical cable mode, soil model, excited vibration, environmental factors, etc test case.

Response 2: The title of the article should be changed to “How the Material Characteristics of Optical Fibers and Soil Influence the Measurement Results of DAS”.

Point 3: The abstract does not provide readers with critical inputs and outcome numeric values. For example, the manuscript's main body examined too little of the cable structure to be included in the abstract.

Response 3: The abstract has been revised accordingly. “Fiber optic distributed acoustic sensing (DAS) technology is widely used in security surveillance and geophysical survey applications. The response of the DAS system to the external vibrations varies with different types of fiber optic cable connections. The mechanism of mutual influence between the physical properties, and the surrounding environment remains unclear. This study proposed a dynamic model of the interaction between the optical cable and the soil, analyzed the impact of the dynamic parameters of the optical cable and soil on the sensitivity of the DAS system, and validated the theoretical analysis through experiments. The findings suggest that augmenting the cable's bending stiffness by 5.5 times and increasing its unit mass by 4.2 times result in a discernible reduction of the system's response to roughly 0.15 times of its initial magnitude. Cables with lower unit mass and bending stiffness are more sensitive to vibration signals. This research provides a foundation for optimizing vibration-enhanced fiber optic cables and broadening the potential usage scenarios for DAS systems.”

Point 4: The conclusion is not supported by the test setup as it is very narrowly focused on selected test assumptions and parameters.

Response 4: The conclusion has been revised accordingly. “The DAS system sensitivity may fluctuate several times due to the influence of the optical cables and their layout methods. We proposed a dynamic model of the interaction between the optical cable and the soil, investigated the relationship between cable mechanical characteristics and DAS sensitivity. The theoretical results indicated that soil stiffness, cable bending stiffness, and cable unit mass are crucial factors that impact the sensitivity of the system. The experimental results indicated that augmenting the cable's bending stiffness by 5.5 times and increasing its unit mass by 4.2 times result in a discernible reduction of the system's response to roughly 0.15 times of its initial magnitude. Cables with lower unit mass and bending stiffness are more sensitive to vibration signals. The present study investigates the impact of cable parameters on the system sensitivity when the cable is considered as an indivisible entity. Future studies will investigate the influence of the in-ternal structure of optical cable on the sensitivity of the system, creating a design system for sensing cable and furnishing a targeted design basis for the application of sensing cable in various fields.”

Point 5: Line 48, there are many commercial custom sensing cables designed for DAS's vibration sensing globally. It is advised for the authors to conduct a more vigorous background study on the type of cable used and use the same or similar structural design for the literature review and the main body of the manuscript.

Response 5: The relevant content has been added to the article. “AP Sensing, Corning [24] and other optical cable manufacturing companies offer the customization of sensing optical cables to achieve a balance between cable thermal conductivity and ruggedness and costs. ”

Point 6: Line 75, is the "Kinetic model of Buried Optical Cable" cited from published work or newly proposed in this manuscript? Additional references should be added to support the theorem/ laws.

Response 6: The dynamic model is an improvement upon the Winkler elastic foundation beam model, and relevant references have been added.

Point 7: Are the assumptions sufficient to model practical seismic waves? Can the authors provide information on such as seismic waves as a use case in the manuscript with reference(s)? If not, the proposed D’Alembert principle, the dynamic balance equation, and the moment balance equation method may be not technically sound for modeling any optical cable.

Response 7: The model can be used to analyze the response of optical cables under seismic waves. Reference [17-19] mentioned that the method of using the elastic foundation beam can be applied to analyze the dynamic displacement or strain response of buried pipelines during earthquakes. This method can be applied as an analogy to study the interaction between optical fiber cables and the surrounding soil.

Point 8: Since the main bulk of the manuscript is on models, the background review should also include past publications on theoretical models.

Response 8: In the introduction, a summary of the theoretical models has been included.

Point 9: Line 152, can the authors kindly provide information on how the typical magnitude values are obtained? Is it from published work or from experiences/ internal data by the authors?

Response 9: The soil parameters were selected based on reference [28], while the relevant parameters for the optical fiber cable and the load were determined according to the conditions set in the subsequent experiments.

Point 10: Line 152, 200, and Section 3 Analysis of parameters affecting DAS sensitivity, a periodic 50 Hz vibration are too simple for analysis for investigating DAS nowadays. Is suggested to use realistic periodic signals for the targeted seismic wave of interest.

Response 10: We utilized a 50Hz sinusoidal vibration exciter as the field test's vibration source. Its distinct vibration characteristics make it suitable for comparing and evaluating the response of different material characteristic optical fibers to vibrations. Additionally, its controllable vibration allows for repetitive testing. In the subsequent experiments, we plan to test a broader range of optical fiber cables with different characteristics under various excitation conditions. This will allow us to gain further insights into how different cables respond to different vibration sources.

Point 11: Line 190, a sum-up table will be useful for readers to glance through the data and results efficiently.

Response 11: The summary table has been added.

Point 12: Line 204, the cross-sectional detailed layout of the optical cables should be depicted in the manuscript. The manufacturer, model information, and specifications of the cables should be provided in the manuscript for a high-quality manuscript.

Response 12: The cross-sectional diagrams of the three types of optical cables have been added to Figure 4. The manufacturer of GYTA53 fiber optic cable is YOFC.

Point 13: Line 208, it is unclear what is the main material in Table 1. As a user of the GYTA52 cable myself, more information on the setup should be provided and justified as to why the current communication optical cable is not practical for seismic wave sensing.

Response 13: The table has been modified, and Figure 4 shows the main material of three cables. Commercial optical cables are typically equipped with metal armor or loose tube structures to minimize stress on the optical fibers, ensuring their longevity. However, this may lead to a decrease in the fiber's sensitivity. The increased use of metal components in the cables results in higher unit mass and bending stiffness, consequently reducing the sensitivity of the optical fibers. According to our subsequent research, the loose tube structure inside the optical cable also has a considerable impact on its sensitivity.

Point 14: Is the 8.00 mm outer diameter cable specification too thin and not practical for buried seismic wave sensing? There is no other structural information inside the manuscript to determine the mechanical properties of the proposed cables' setup. Even the types of sand and how it is set up in the pit will influence the results for example.

Response 14: In this study, three types of optical cables were chosen to create significant differences in unit weight and bending stiffness. The purpose was to validate the theoretical results presented in the paper, rather than for practical engineering applications. Subsequently, based on this theory and experimental results, we will further optimize and design the sensing optical cables for on-site applications.

Point 15: Figure 6. Cable 2 and 3 seem to have stable amplitudes while cable 1 varies a bit. Are there other times when there are large variations? Any reason for the variations?

Response 15: The data obtained from measurements on Optical Cable 2 and Optical Cable 3 also exhibit fluctuations, albeit less prominently compared to the significant amplitude variations observed in Optical Cable 1. The reason for these fluctuations could be attributed to incomplete coupling between the vibration source and the surrounding soil. This can be deduced from the spectral analysis of the measurement results, which reveals the presence of not only the dominant frequency of 50Hz but also higher harmonics such as 100Hz and other noise frequencies. The most plausible explanation for this phenomenon is the development of gaps between the vibrating source and the soil due to prolonged vibrations of the exciter. These gaps are believed to induce the appearance of other frequencies, ultimately resulting in signal fluctuations.

Reviewer 3 Report

The article is well described and all the measurements for the two tests indicate that soil stiffness, cable bending stiffness, cable length, and cable unit mass are crucial factors that impact the sensitivity of the system. This novel application that is demostrated, help to investigate the relationship between the strain response of the optical fiber and system parameters such as the bending stiffness,linear density, and length of the cable. The colclusions will be improved by adding some conclusions supported by the results of Test 1 aims that investigate the response of cables under different soil stiffness, with cables being buried to the same length and Test 2 that shows the response of cables with various buried lengths under a uniform soil stiffness, and a common excitation source.

A similiar approach is for automatic vehicle detection and counting by processing data acquired using a phase-sensitive optical time-domain reflectometer (?-OTDR) distributed optical fiber sensor, has been experimentally tested by performing ?-OTDR measurements along a telecommunication fiber cable, underground of normal traffic. [Ester Catalano, Agnese Coscetta, Enis Cerri, Nunzio Cennamo, Luigi Zeni, and A. Minardo, "Automatic traffic monitoring by ϕ-OTDR data and Hough transform in a real-field environment," Appl. Opt. 60, 3579-3584 (2021)].

Author Response

The article is well described and all the measurements for the two tests indicate that soil stiffness, cable bending stiffness, cable length, and cable unit mass are crucial factors that impact the sensitivity of the system. This novel application that is demostrated, help to investigate the relationship between the strain response of the optical fiber and system parameters such as the bending stiffness,linear density, and length of the cable. The colclusions will be improved by adding some conclusions supported by the results of Test 1 aims that investigate the response of cables under different soil stiffness, with cables being buried to the same length and Test 2 that shows the response of cables with various buried lengths under a uniform soil stiffness, and a common excitation source.

Point 1:A similiar approach is for automatic vehicle detection and counting by processing data acquired using a phase-sensitive optical time-domain reflectometer (?-OTDR) distributed optical fiber sensor, has been experimentally tested by performing ?-OTDR measurements along a telecommunication fiber cable, underground of normal traffic. [Ester Catalano, Agnese Coscetta, Enis Cerri, Nunzio Cennamo, Luigi Zeni, and A. Minardo, "Automatic traffic monitoring by ϕ-OTDR data and Hough transform in a real-field environment," Appl. Opt. 60, 3579-3584 (2021)].

Response 1: The references have been added to the introduction section.

Reviewer 4 Report

The authors presented a rather interesting work having the timely theme. They have provided a dynamic model of the interaction between the optical cable and the soil, analyzed the impact of the dynamic parameters of the optical cable and soil on the sensitivity of the DAS system, and validated the theoretical analysis through experiments. They have claimed the results which indicate that soil stiffness, cable length, and cable unit weight are key factors influencing cable response sensitivity. I believe that this manuscript contains interesting practical knowledge and will be of interest to Sensors readers. However, before publication, I would draw the attention of the authors to some remarks. I believe that after eliminating these minor shortcomings, the article will be ready for publication:

1. The abstract does not contain quantitative results, which are given in the Discussion section. In my opinion, providing specific values of physical quantities will give a clearer idea of the results.

2. I think that the list of references in the Introduction need to be expanded. In fact, there are quite a few methods for increasing the sensitivity and signal-to-noise ratio of DAS systems. Some of these may be alternatives to the proposed one, some may work together. I consider it appropriate to mention not only cable design modifications, but also modifications to the design of the optical fiber[http://dx.doi.org/10.1109/JLT.2023.3281136], [http://dx.doi.org/10.1109/ACCESS.2021.3105334]; updating the system hardware [http://dx.doi.org/10.3390/s22239482], [http://dx.doi.org/10.3389/fphy.2023.1196067]; and digital signal processing methods [http://dx.doi.org/10.3390/a16050217], [http://dx.doi.org/10.1109/JLT.2023.3273268].

3. If you use a phase-sensitive reflectometer of your own design, since the experiment should be repeatable, you need to provide the setup drawing. If this is not possible (say, due to commercial secret), then it is desirable to give more detailed performance characteristics: the bandwidth of the laser, its optical power, the pulse durations used, how the signal was amplified, filtered and detected, etc.

4. Of course, DAS systems are mainly used to monitor fairly long lines. But since the laser parameters are not specified in this study, I would ask the authors to make sure that they get the same results for probing the entire line from the other side (in the opposite direction). Could these different responses on the three cables be also related to the degradation of the sensing parameters towards the end of the line?

5. In the table where the cable parameters are given, it would be great to also indicate the types of these fibers. Can you provide drawings of cable cross-sections?

6. Despite the fact that the English language is quite good and the text is easy to read, there are quite a few typos in the manuscript that need to be eliminated.

7. The same that I stated for the abstract above, is largely true for the conclusion: in the final part of the manuscript, I would ask the authors to duplicate the quantitative value of the physical quantities they obtained.

8. Figure 1 appears before it's mentioned in the text. Is this okay?

Despite the fact that the English language is quite good and the text is easy to read, there are quite a few typos in the manuscript that need to be eliminated

Author Response

Response to Reviewer 4 Comments

The authors presented a rather interesting work having the timely theme. They have provided a dynamic model of the interaction between the optical cable and the soil, analyzed the impact of the dynamic parameters of the optical cable and soil on the sensitivity of the DAS system, and validated the theoretical analysis through experiments. They have claimed the results which indicate that soil stiffness, cable length, and cable unit weight are key factors influencing cable response sensitivity. I believe that this manuscript contains interesting practical knowledge and will be of interest to Sensors readers. However, before publication, I would draw the attention of the authors to some remarks. I believe that after eliminating these minor shortcomings, the article will be ready for publication:

Point 1: The abstract does not contain quantitative results, which are given in the Discussion section. In my opinion, providing specific values of physical quantities will give a clearer idea of the results.

Response 1: The abstract has been revised accordingly. “Fiber optic distributed acoustic sensing (DAS) technology is widely used in security surveillance and geophysical survey applications. The response of the DAS system to the external vibrations varies with different types of fiber optic cable connections. The mechanism of mutual influence between the physical properties, and the surrounding environment remains unclear. This study proposed a dynamic model of the interaction between the optical cable and the soil, analyzed the impact of the dynamic parameters of the optical cable and soil on the sensitivity of the DAS system, and validated the theoretical analysis through experiments. The findings suggest that augmenting the cable's bending stiffness by 5.5 times and increasing its unit mass by 4.2 times result in a discernible reduction of the system's response to roughly 0.15 times of its initial magnitude. Cables with lower unit mass and bending stiffness are more sensitive to vibration signals. This research provides a foundation for optimizing vibration-enhanced fiber optic cables and broadening the potential usage scenarios for DAS systems.”

Point 2: I think that the list of references in the Introduction need to be expanded. In fact, there are quite a few methods for increasing the sensitivity and signal-to-noise ratio of DAS systems. Some of these may be alternatives to the proposed one, some may work together. I consider it appropriate to mention not only cable design modifications, but also modifications to the design of the optical fiber[http://dx.doi.org/10.1109/JLT.2023.3281136], [http://dx.doi.org/10.1109/ACCESS.2021.3105334]; updating the system hardware [http://dx.doi.org/10.3390/s22239482], [http://dx.doi.org/10.3389/fphy.2023.1196067]; and digital signal processing methods [http://dx.doi.org/10.3390/a16050217], [http://dx.doi.org/10.1109/JLT.2023.3273268].

Response 2: The references have been added to the introduction section.

Point 3: If you use a phase-sensitive reflectometer of your own design, since the experiment should be repeatable, you need to provide the setup drawing. If this is not possible (say, due to commercial secret), then it is desirable to give more detailed performance characteristics: the bandwidth of the laser, its optical power, the pulse durations used, how the signal was amplified, filtered and detected, etc.

Response 3: Phase demodulation technique of DAS is based on 3×3 coupler. The pulse width of the DAS system is 50ns, the optical source has a power of 20mW, the sampling rate is 20000Hz, the gauge length is 5m. The relevant content in the paper have been revised accordingly.

Point 4: Of course, DAS systems are mainly used to monitor fairly long lines. But since the laser parameters are not specified in this study, I would ask the authors to make sure that they get the same results for probing the entire line from the other side (in the opposite direction). Could these different responses on the three cables be also related to the degradation of the sensing parameters towards the end of the line?

Response 4: In the manuscript, we positioned optical cable 3 at the farthest distance from DAS. However, during the experiment, we tested optical cable 3 both as the nearest and the farthest, and the measurements consistently indicated optical cable 3 to be the weakest. During the experiment, we also attempted to minimize fusion splicing of optical fibers and increase the bending radius of the fibers to reduce the impact of attenuation on measurement results as much as possible.

Point 5: In the table where the cable parameters are given, it would be great to also indicate the types of these fibers. Can you provide drawings of cable cross-sections?

Response 5: The cross-sectional diagrams of the three types of optical cables have been added to Figure 4.

Point 6: Despite the fact that the English language is quite good and the text is easy to read, there are quite a few typos in the manuscript that need to be eliminated.

Response 6: Typographical errors have been checked and corrected.

Point 7: The same that I stated for the abstract above, is largely true for the conclusion: in the final part of the manuscript, I would ask the authors to duplicate the quantitative value of the physical quantities they obtained.

Response 7: The conclusion has been revised accordingly. “The DAS system sensitivity may fluctuate several times due to the influence of the optical cables and their layout methods. We proposed a dynamic model of the interaction between the optical cable and the soil, investigated the relationship between cable mechanical characteristics and DAS sensitivity. The theoretical results indicated that soil stiffness, cable bending stiffness, and cable unit mass are crucial factors that impact the sensitivity of the system. The experimental results indicated that augmenting the cable's bending stiffness by 5.5 times and increasing its unit mass by 4.2 times result in a discernible reduction of the system's response to roughly 0.15 times of its initial magnitude. Cables with lower unit mass and bending stiffness are more sensitive to vibration signals. The present study investigates the impact of cable parameters on the system sensitivity when the cable is considered as an indivisible entity. Future studies will investigate the influence of the in-ternal structure of optical cable on the sensitivity of the system, creating a design system for sensing cable and furnishing a targeted design basis for the application of sensing cable in various fields.”

Point 8: Figure 1 appears before it's mentioned in the text. Is this okay?

Response 8: The position of Figure 1 has been modified.

Round 2

Reviewer 1 Report

After a careful reading of the manuscript, in general, the authors answered the questions posed and produced significant improvements regarding the understanding of the work performed.

Author Response

Thank you for reviewing the manuscript. We appreciate your time and consideration. 

Reviewer 2 Report

This is the second review report for the manuscript titled "Research on the Vibration response of Fiber Optic Cable in Soil for DAS Measurement" by Ke Jiang, et al. The manuscript title has been revised to "How the Material Characteristics of Optical Fibers and Soil Influence the Measurement Results of DAS" in the new version.

Further queries:

Point 5: Line 48, there are many commercial custom sensing cables designed for DAS's vibration sensing globally. It is advised for the authors to conduct a more vigorous background study on the type of cable used and use the same or similar structural design for the literature review and the main body of the manuscript.

Response 5: The relevant content has been added to the article. “AP Sensing, Corning [24] and other optical cable manufacturing companies offer the customization of sensing optical cables to achieve a balance between cable thermal conductivity and ruggedness and costs. ”

1) The authors have included a relevant paper on analysing the different types of cross sections cables. However, the article and the manuscript itself still do not cover sufficient background review. Explicitly on cable design like diagrams of the cables' cross sections. There are also many existing products (cable design) for the purpose of distributed acoustic sensing.

2) Another point is that since the manuscript cited reference [24] and that paper demonstrated the design 'distributed sensor cable (DSC)' has higher sensitivity than communication optical fiber cable (which has a similar design to GYTA53), why is the manuscript still using an inferior design for the experiment?

Point 12: Line 204, the cross-sectional detailed layout of the optical cables should be depicted in the manuscript. The manufacturer, model information, and specifications of the cables should be provided in the manuscript for a high-quality manuscript.

Response 12: The cross-sectional diagrams of the three types of optical cables have been added to Figure 4. The manufacturer of GYTA53 fiber optic cable is YOFC.

For completeness of the research:

3) The manuscript should include the manufacturer name of the GYTA53 cable. 

4) How about the information/specifications for aluminum, rubber (TPU), etc.? 

5) How is the tight buffered cable affixed to the test cables? For example, information by means of epoxy, affix methods, etc.?

Point 13: Line 208, it is unclear what is the main material in Table 1. As a user of the GYTA52 cable myself, more information on the setup should be provided and justified as to why the current communication optical cable is not practical for seismic wave sensing.

Response 13 "The table has been modified, and Figure 4 shows the main material of three cables. Commercial optical cables are typically equipped with metal armor or loose tube structures to minimize stress on the optical fibers, ensuring their longevity. However, this may lead to a decrease in the fiber's sensitivity. The increased use of metal components in the cables results in higher unit mass and bending stiffness, consequently reducing the sensitivity of the optical fibers. According to our subsequent research, the loose tube structure inside the optical cable also has a considerable impact on its sensitivity."

My apologize for typo on 'GYTA53' cable in review report 1. 

5) Have the authors considered other aspects such as the filling compound inside the cable design that dampens vibration signals? Further discussion should be added to the manuscript for completeness and avoid solely replying reviewer's comment only. 

Point 14: Is the 8.00 mm outer diameter cable specification too thin and not practical for buried seismic wave sensing? There is no other structural information inside the manuscript to determine the mechanical properties of the proposed cables' setup. Even the types of sand and how it is set up in the pit will influence the results for example.

Response 14: In this study, three types of optical cables were chosen to create significant differences in unit weight and bending stiffness. The purpose was to validate the theoretical results presented in the paper, rather than for practical engineering applications. Subsequently, based on this theory and experimental results, we will further optimize and design the sensing optical cables for on-site applications.

6) The manuscript has provided some results pertaining to the cable mass and bending stiffness with vibration sensing. However, the application for this research is clearly for buried usage and this application approach should be addressed as part of the criteria. Perhaps some discussion on the reference [24] DSC's cable design with cable 2 design could be linked and explained. 

I hope my comments will improve the authors' manuscript in a good way. 

Author Response

Response to Reviewer 2 Comments

Point 5: Line 48, there are many commercial custom sensing cables designed for DAS's vibration sensing globally. It is advised for the authors to conduct a more vigorous background study on the type of cable used and use the same or similar structural design for the literature review and the main body of the manuscript.

Response 5: The relevant content has been added to the article. “AP Sensing, Corning [24] and other optical cable manufacturing companies offer the customization of sensing optical cables to achieve a balance between cable thermal conductivity and ruggedness and costs. ”

Point 1: The authors have included a relevant paper on analysing the different types of cross sections cables. However, the article and the manuscript itself still do not cover sufficient background review. Explicitly on cable design like diagrams of the cables' cross sections. There are also many existing products (cable design) for the purpose of distributed acoustic sensing.

Response 1: The cross-section table of the optical fiber cable has been incorporated in the background review.

Point 2: Another point is that since the manuscript cited reference [24] and that paper demonstrated the design 'distributed sensor cable (DSC)' has higher sensitivity than communication optical fiber cable (which has a similar design to GYTA53), why is the manuscript still using an inferior design for the experiment?

Response 2: Firstly, GYTA 53 is the most commonly used standard buried communication optical cable, serving as a benchmark for our comparative testing. Secondly, this study focuses on the impact of cable characteristics on DAS measurement results. Therefore, it is important to minimize the influence of other factors on the tests, such as the internal structure and overall shape of the optical cable. The DSC (butterfly-shaped) cable mentioned in reference [24] has varying bending stiffness in the X and Y directions, which could introduce additional effects on the measurement results. Hence, in this study, we have selected GYTA53 and two other simulated cables that have similar diameters and shapes, but with differences in unit mass and bending stiffness for comparison.

Point 12: Line 204, the cross-sectional detailed layout of the optical cables should be depicted in the manuscript. The manufacturer, model information, and specifications of the cables should be provided in the manuscript for a high-quality manuscript.

Response 12: The cross-sectional diagrams of the three types of optical cables have been added to Figure 4. The manufacturer of GYTA53 fiber optic cable is YOFC.

For completeness of the research:

Point 3: The manuscript should include the manufacturer name of the GYTA53 cable.

Response 3: The relevant content has been revised accordingly.

Point 4: How about the information/specifications for aluminum, rubber (TPU), etc.?

Response 4: The relevant content has been revised accordingly. ”The density and elastic modulus of aluminum and TPU are 2.72g/cm3, 79Gpa and 1.17g/cm3, 32Mpa respectively.”

Point 5: How is the tight buffered cable affixed to the test cables? For example, information by means of epoxy, affix methods, etc.?

Response 5: The relevant content has been revised accordingly. “A 0.9mm tight-buffered cable is fixed to the outer surface of three test cables employing the utilization of heat-shrink tubing.”

Point 6: Have the authors considered other aspects such as the filling compound inside the cable design that dampens vibration signals? Further discussion should be added to the manuscript for completeness and avoid solely replying reviewer's comment only.

Response 6: The relevant content has been revised accordingly. ”Future studies will investigate the influence of the internal structure of optical cable, such as the stretching window, twisting angle, grease filling of optical units, as well as the inner structural design of reinforcing elements”.

Point 7: The manuscript has provided some results pertaining to the cable mass and bending stiffness with vibration sensing. However, the application for this research is clearly for buried usage and this application approach should be addressed as part of the criteria. Perhaps some discussion on the reference [24] DSC's cable design with cable 2 design could be linked and explained.

Response 7: The relevant content has been revised accordingly. “There are typically two methods for laying optical cables within soil, direct burial, and duct installation. This article only discusses the scenario of direct burial. The measurements from reference [25] indicate that the measured signal when directly buried is approximately 10 dB higher compared to duct installation. We will also delve into the dynamic model of cable deployment within duct and analysis the influencing factors. Additionally, the sensor cables designed in references [24] [25] all share char-acteristics of being lightweight, having low bending stiffness, and tight fitting. This further validates the reliability of the findings presented in this article”.
